

# Climate change and tree cover loss affect the habitat suitability of *Cedrela angustifolia*: evaluating climate vulnerability and conservation in Andean montane forests

Fressia N. Ames-Martínez[1], Ivan Capcha Romero[2], Anthony Guerra[2], Janet Gaby Inga Guillen[3], Harold Rusbelth Quispe-Melgar[4,5,6], Esteban Galeano[7] and Ernesto C. Rodríguez-Ramírez[8]

[1] Laboratorio de Biotecnología y Biología Molecular, Universidad Continental, Huancayo, Junin, Peru
[2] Facultad de Ciencias Forestales y del Ambiente, Universidad Nacional del Centro del Perú, Huancayo, Junin, Peru
[3] Laboratorio de la Anatomía e Identificación de la Madera, Universidad Continental, Huancayo, Junin, Peru
[4] Programa de Ecología y Diversidad, Asociación ANDINUS, Huancayo, Junin, Peru
[5] Facultad de Ciencias de la Salud, Universidad Continental, Huancayo, Junin, Peru
[6] Estación Experimental Agraria Santa Ana, Instituto Nacional de Innovación Agraria, Huancayo, Junin, Peru
[7] Department of Forestry, College of Forest Resources, Mississippi State University, Mississipi, United States of America
[8] Laboratorio de Dendrocronología, Universidad Continental, Huancayo, Junin, Peru

Corresponding author
Fressia N. Ames-Martínez,
fressiames@gmail.com

## ABSTRACT

**Background.** Because of illegal logging, habitat fragmentation, and high value timber Andean montane forest *Cedrela* species (such as *Cedrela angustifolia*), is endangered in Central and South America. Studying the effects of climate change and tree cover loss on the distribution of *C. angustifolia* will help us to understand the climatic and ecological sensitivity of this species and suggest conservation and restoration strategies.

**Methods.** Using ecological niche modeling with two algorithms (maximum entropy (MaxEnt) and Random Forest) under the ecological niche conservatism approach, we generated 16,920 models with different combinations of variables and parameters. We identified suitable areas for *C. angustifolia* trees under present and future climate scenarios (2040, 2070, and 2100 with SSP 3-7.0 and SSP 5-8.5), tree cover loss, and variables linked to soil and topography.

**Results.** Our results demonstrated 10 environmental variables with high percentage contributions and permutation importance; for example, precipitation seasonality exhibited the highest contribution to the current and future distribution of *Cedrela angustifolia*. The potential present distribution was estimated as 13,080 km$^2$ with tree cover loss and 16,148.5 km$^2$ without tree cover loss. From 2040 to 2100 the species distribution will decrease (from 22.16% to 36.88% with tree cover loss variation). The results indicated that Bolivia displayed higher habitat suitability than Ecuador, Peru, and Argentina. Finally, we recommend developing conservation management strategies that consider both protected and unprotected areas as well as the impact of land-use changes to improve the persistence of *C. angustifolia* in the future.

## INTRODUCTION

Climate change displays a significant impact on the decline of Andean tree populations, and even on the extinction of range-restricted species (*Tejedor Garavito et al., 2015*; *Urrutia & Vuille, 2009*). Likewise, anthropic activities such as habitat destruction and illegal logging can lead to the extinction of threatened species (*Pievani, 2014*). Climate variability influences autecological processes and environmental fluctuations (*Anderson & Song, 2020*). For example, variations in temperature and precipitation affect the wood anatomical plasticity, phenology, climatic resilience, geographic range, productivity, and disruption of inter- and intraspecific relationships (*Araújo & Rahbek, 2006*; *Fonti et al., 2010*; *Piao et al., 2019*). Therefore, understanding the fate of tree species in response to climate change is essential for formulating effective strategies to acclimate or mitigate the impacts of climate change on both natural and anthropic activities (*Urrutia & Vuille, 2009*).

Bioclimatic niche modeling shows various applications in the paradigm of climate change in vulnerable areas, and researchers appreciate its anthropic use (*Urrutia & Vuille, 2009*). The incorporation of ecological data into niche modeling highlights the links between ecological processes and climate change dynamics. A thorough understanding of the montane ecological niche, and its adaptive responses to projected climate change scenarios will improve the identification and prioritization of conservation strategies within Andean ecosystems. This will encourage sustainable resource management, particularly in fragile ecosystems (*Llambí & Garcés, 2020*; *Urrutia & Vuille, 2009*).

Andean montane forests (AMFs; *Bush, Hanselman & Hooghiemstra, 2007*) constitute a significant part of the tropical Andean biodiversity hotspot (*Myers et al., 2000*). AMFs provide a stable balance within the community of organisms in which genetic, species and ecosystem diversity remain subject to gradual change through natural succession, high species richness, and ecosystem services to both the high- and low moisture areas of the Andes (*Cuesta, Peralvo & Valarezo, 2009*; *Myers et al., 2000*). Forest fires, tree cover loss, and climate change have influenced major changes in montane ecosystems over the last century, including ecosystem fragmentation and loss of diversity (*Feeley & Silman, 2010*; *Gaglio et al., 2017*; *Rolando et al., 2017*). Hence, the establishment of sustainable management strategies for threatened AMF tree species is of multinational interest. It is therefore important to understand the effects of climate change on these species.

The arboreal genus *Cedrela* L. (Meliaceae) is a protected tree species (Convention on International Trade in Endangered Species (CITES) and International Union for Conservation of Nature (IUCN); *Pennington & Muellner, 2010*), comprising 19 species widely distributed from North America to the tropical mountains of South America, where it inhabits steep ravines (*Köcke et al., 2015*; *Muellner et al., 2010*; *Palacios, Santiana & Iglesias, 2019*; *Pennington & Muellner, 2010*). *Cedrela angustifolia* Moc. & Sessé ex DC. (VU; as indicated by *Hills (2021)* in the IUCN Red List; http://www.iucnredlist.org/). This

species is ecologically important as it is a pioneer and co-dominant species associated with *Oreopanax*, *Podocarpus*, and *Weinmania* (*Pennington & Muellner, 2010*). Additionally, they play a key role in providing essential ecosystem services, including firewood, timber, particle boards, furniture, flooring veneer, and railway tires (*Pennington & Muellner, 2010*; *SERFOR, 2020*).

*C. angustifolia* is a pioneer species, that thrives in disturbed areas and contributes to AMF regeneration by providing shade and habitat for other plants, fungi and animals (*Reynel & Pennington, 1989*). Likewise, these trees enhance biodiversity by supporting various organisms within tropical and subtropical ecosystems, and the flowers are visited by small bees and butterflies, although it is not yet clear whether these insects are legitimate pollinators, and the seeds are dispersed by the wind (*SERFOR, 2020*). Nevertheless, their populations are threatened by overexploitation and habitat loss, and conservation efforts are needed. Finally, human communities that rely on *C. angustifolia* for construction and furniture face reduced income because of dwindling supplies, as overexploitation has resulted in fewer mature trees available for harvest (*Reynel & Pennington, 1989*; *SERFOR, 2020*).

Species distribution models (SDM) are an essential tool for identifying climatic refugia in areas with changing abiotic conditions, which is crucial for understanding the species' ecological niches and identifying potential habitats, especially in the context of changing environments. Using occurrence and environmental data, they frequently used SDMs as tools to estimate the extent of a species' range in the future or past (*Peterson et al., 2011*). Likewise, tropical forests are vulnerable to logging, deforestation, and loss of tree cover, particularly in regions previously characterized by colder climates (*Gaglio et al., 2017*; *Sarmiento, 2002*).

Conversion of forests to agricultural land, particularly for livestock and avocado or granadilla crops, is a poorly regulated logging activity, often driven by global demand for timber, which further exacerbates forest loss (*Bax & Francesconi, 2018*; *Cuenca, Arriagada & Echeverría, 2016*). These changes are linked to the acceleration of atmospheric $CO_2$ concentrations due to changes in energy, mass and momentum exchange (*Friedlingstein et al., 1999*). Therefore, it is necessary to understand how climate change and forest cover loss affect the distribution of *C. angustifolia*.

In this study, we hypothesized that the most suitable habitat for *C. angustifolia* would be negatively affected by climate change and tree cover loss by 2100. Therefore, it is necessary to determine the effectiveness of protected areas and suitable sites for restoration efforts and ecological refugia to maintain viable populations in South America. The findings of this study can be used to establish natural reserves and conservation areas for *C. angustifolia* and to provide information relevant to IUCN Red List status. Our main aims were to: (1) identify the environmental variables responsible for the present and future potential distribution of *C. angustifolia*; (2) assess the effect of climate sensitivity and tree cover loss on *C. angustifolia* by comparing the present and future potential distribution; (3) evaluate the effect of tree cover loss on the potential distribution of *C. angustifolia* in present and future scenarios up to 2100; and (4) determine the potential refugia for climate change and conservation of *C. angustifolia*, relating natural protected areas (NPAs), land use change,

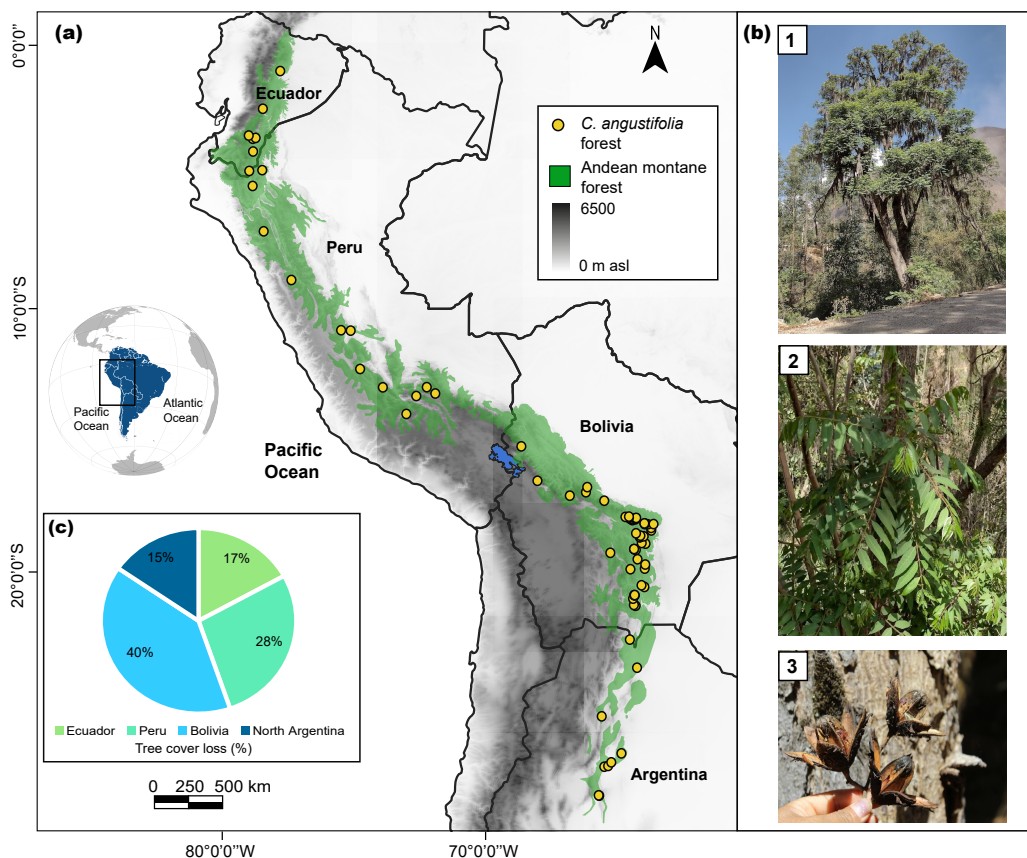

**Figure 1** **Study area for *Cedrela angustifolia* climate sensitivity analysis in the Andean montane forests.** (A) Current distribution of *C. angustifolia* forest in Ecuador, Peru, Bolivia, and Argentina (realized by Fressia Nathalie Ames Martínez); (B) *C. angustifolia* dasometric features: 1 = tree, 2 = leaves, 3 = fruit; and (C) tree cover loss of this species categorized by country, corresponding to the year 2021 (*Hansen et al., 2013*; *Harris et al., 2021*). Photos by Fressia Nathalie Ames Martínez and base map for countries was obtained from DIVA-GIS (2017).

present, and future potential distribution data. This study aimed to evaluate the response of *C. angustifolia* to climate pressures and to provide recommendations and conservation strategies that will have significant impact on countries in a mega-diverse tropical region.

## MATERIAL AND METHODS

### Study area

The study area covered the Andean montane forest region (Ecuador, Peru, Bolivia, and Argentina; Fig. 1), with elevations ranging from ≈1,800 to 3,300 m asl, as recorded by *Pennington & Muellner (2010)*. The study was delimited by Tungurahua Province, Ecuador, in the north (1°S), and Catamarca Province, Argentina, in the south (28°S). The Servicio Nacional Forestal y de Fauna Silvestre-SERFOR approved the research outside Protected Natural Areas through the General Management Resolution RDG 007-2020-MINAGRI-SERFOR-DGGSPFFS.

## Data sampling

We obtained occurrence data for *Cedrela angustifolia* from the Tropicos database (https://tropicos.org/home; *Missouri Botanical Garden, 2022*), the Global Biodiversity Information Facility database (https://gbif.org/; *GBIF Secretariat, 2022*), and scientific publications (*i.e., Inza et al., 2012; Paredes-Villanueva, López & Navarro Cerrillo, 2016; Pennington & Muellner, 2010; Wong & Reynel, 2021*) totaling 310 occurrences. In addition, we checked and excluded incorrectly georeferenced points, outliers and herbarium accessions with missing location data. We also used geographical distances to narrow our datasets. We clustered occurrences to a grid size of ~one km$^2$ based on the environmental heterogeneity of the ecosystems in which *C. angustifolia* occurs, such as montane forests and inter-Andean valleys, to filter out closely duplicated records. Finally, 104 records were obtained from Ecuador, Peru, Bolivia, and Argentina (89 GBIF and Tropicos.org and 15 scientific studies).

## Data preparation

We used 39 environmental variables: 22 bioclimatic variables for the present and future periods, elevation, 11 soil raster layers, one NDVI raster, and four forest change data rasters (Table S1). Under the assumption that it influences niche conservatism (*Peterson et al., 2011*), the climate sensitivity of *C. angustifolia* is related to its present and future distributions (2011–2040, 2041–2070, and 2071–2100 years). We obtained raster model bioclimatic data from the CHELSA database with a spatial resolution of 1.0 km$^2$ (https://chelsa-climate.org/; *Karger et al., 2022*) for the present and future periods. In the present and future models, we included 19 raster model bioclimatic variables and potential evapotranspiration (PET), climate moisture index (CMI), and near-surface relative humidity (HURS) (Table 1).

We used 'Coupled Model Intercomparison Project Phase 6' (CMIP6), which is part of the 'Working Group on Coupled Modelling' (WGCM), to assess projections of bioclimatic variables to three different times in the future (*Eyring et al., 2016*). We considered the model projections because the models are given a common set of future concentrations of greenhouse gases, aerosols, and other climate forcing to project what might happen in the future (*Eyring et al., 2016*).

We selected two future climate scenarios (shared socioeconomic pathways; SSP 3-7.0 and 5-8.5) for the five models to derive future climate projections (Table S1). We assume that SSP 3-7.0 is a less chaotic scenario because it represents a smaller reduction in global greenhouse gas concentrations by 2100 than SSP 5-8.5, which is the highest carbon emissions scenario and the most pessimistic view of the future. We considered a digital elevation model (DEM) variable (*Karger et al., 2022*) to assess the elevation effects.

In addition, we obtained 11 soil raster layers from SoilGrid (https://soilgrids.org/; *Batjes, Ribeiro & Van Oostrum, 2020*) with a spatial resolution of 250 m. Moreover, we obtained global forest change data (land cover, tree cover loss, loss and gain of forest raster layers from 2000 to 2022) from Global Forest Watch, with approximately 30 m per pixel (https://www.globalforestwatch.org/map/; *Hansen et al., 2013*). Furthermore, we used the Normalized Difference Vegetation Index (NDVI), which indicates the vegetation coverage

Ames-Martínez et al. (2025), *PeerJ*, DOI 10.7717/peerj.18799

**Table 1 Information about the all environmental variables of present and future models.** The table shows the type variables, time, model names, spatial resolutions, periods, and variables, literature sources. The bold type face indicates that they are selected for the ensemble model.

| Type | Time | Model name | Spatial resolution | Period | | Variables | Source |
|------|------|-----------|--------------------|--------|---|-----------|--------|
| Climatic | Present | Present | 1 km² | 1980–2010 | BIO1 | Annual mean temperature | CHELSA (*Karger et al., 2022*) |
| | | | | | BIO2 | Mean diurnal range | |
| | | | | | **BIO3** | **Isothermality** | |
| | | GFDL-ESM4 | | | **BIO4** | **Temperature seasonality** | |
| | | | | | BIO5 | Max temperature of warmest month | |
| | | | | | **BIO6** | **Min temperature of coldest month** | |
| | | | | | BIO7 | Temperature annual range | |
| | | **IPSL-CM6A-LR** | | | BIO8 | Mean Temperature of wettest quarter | |
| | | | | | BIO9 | Mean temperature of driest quarter | |
| | | | | | **BIO10** | **Mean temperature of warmest quarter** | |
| | Future (SSP 3-7.0 and 5-8.5 scenarios) | | | 2011–2040 2041–2070 2071–2100 | BIO11 | Mean temperature of coldest quarter | |
| | | MPI-ESM1-2-HR | | | BIO12 | Annual precipitation | |
| | | | | | BIO13 | Precipitation of wettest month | |
| | | | | | BIO14 | Precipitation of driest month | |
| | | | | | **BIO15** | **Precipitation seasonality** | |
| | | MRI-ESM2-0 | | | BIO16 | Precipitation of wettest quarter | |
| | | | | | BIO17 | Precipitation of driest quarter | |
| | | | | | **BIO18** | **Precipitation of warmest quarter** | |
| | | | | | **BIO19** | **Precipitation of coldest quarter** | |
| | | | | | PET | potential evapotranspiration | |
| | | UKESM1-0-LL | | | **CMI** | **climate moisture index** | |
| | | | | | HURS | near surface relative humidity | |
| Topographic | | | | | **ALT** | **Altitude** | |

Ames-Martínez et al. (2025), *PeerJ*, DOI 10.7717/peerj.18799

**Table 1** (*continued*)

| Type | Time | Model name | Spatial resolution | Period | Variables | | Source |
|------|------|-----------|-------------------|--------|-----------|---|--------|
| Edaphic | Present | Soil | 250 m | 2000–2022 | BD | Bulk density | *Batjes, Ribeiro & Van Oostrum (2020)* |
| | | | | | CEC | Cation exchange capacity | |
| | | | | | **CC** | **Clay content** | |
| | | | | | CF | Coarse Fragment | |
| | | | | | N | Nitrogen | |
| | | | | | **OCD** | **Organic carbon density** | |
| | | | | | PH | pH water | |
| | | | | | S | Sand | |
| | | | | | **SILT** | **Silt** | |
| | | | | | SOC | Soil organic carbon | |
| | | | | | **SOCS** | **Soil organic carbon stock** | |
| Tree forest change | | Global forest change | 30 m | 2000–2022 | LC | Land cover | *Hansen et al. (2013)* |
| | | | | | **TCL** | **Tree cover loss** | |
| | | | | | **LF** | **Forest loss** | |
| | | | | | GF | Gain forest | |
| Vegetal | | NDVI | 1 km$^2$ | 2000–2022 | **NDVI** | **Normalized Difference Vegetation Index** | Earth explorer |

and spatial distribution of vegetation, calculated as the annual mean NDVI for 2000–2020, using Earth Explorer data with one km² spatial resolution (https://earthexplorer.usgs.gov/).

For present and future models, we assumed that the forest loss, soil and NDVI layers were constant to determine the climate effect on vegetation loss and soil characteristics. Finally, we used the *resample* function so that all environmental variables had a resolution of ~one km², so we used the bioclimatic variables as reference values (Y) and the soil, topography, and forest change layers as the layers to be resampled (X) considering a bilinear interpolation, which were included in the *raster* package (*Hijmans, 2023*) in R software v. 4.3.1 (*R Core Team, 2022*).

## Data analysis

We determined the appropriate selection of environmental variables to perform statistical and ecological parameter analyses of suitable models, according to *Ames-Martínez et al. (2022)* (Fig. 2). We selected variables with multicollinearity based on measures, such as principal component analysis (PCA), to identify climatic variables potentially important in the geographic distribution of *C. angustifolia*. We utilized the entire environmental matrix and we selected the variables that were present in the first two principal components (*Estrada-Peña et al., 2013*) with eigenvalues > 1 and percentage of variance explained 53% and 31% of the variation respectively, for a total of 84%. In addition, we assessed the collinearity of the covariates with the variance inflation factor (VIF) in the pre-selected matrix of the PCA analysis, we sequentially excluded the one with the greatest VIF sequentially until all remaining predictors had VIFs < 10 (*Cobos et al., 2019a*). Furthermore, we selected variables with pairwise Pearson's correlation < 0.7 from the VIF analysis matrix to be used for the species distribution model. Our approach involves the use of *sdm* (*calibration* function; *Naimi & Araújo, 2016*), *fuzzySim* (*corSelect* function; *Barbosa, 2015*), *regclass* (*VIF* function; *Petrie, 2020*), *FactoMineR* (*PCA* function; *Husson et al., 2023*), and *virtual species* packages (*removeCollinearity* function; *Leroy et al., 2019*).

Finally, after variable selection, we selected 16 environmental variables: isothermality (BIO3), temperature seasonality (BIO4), minimum temperature of the coldest month (BIO6), mean temperature of the warmest quarter (BIO10), precipitation seasonality (BIO15), precipitation of the warmest quarter (BIO18), precipitation of the coldest quarter (BIO19), climate moisture index (CMI), altitude (ALT), soil organic carbon stock (SOCS), organic carbon density (OCD), silt content (SILT), clay content (CC), deforestation (DEF), forest loss (LF), and NDVI. For the analysis of tree cover loss, we used the forest loss raster in some models and other models without this raster to compare forest loss in the present and future periods, assuming that the same current rate of forest loss will continue into the future.

## Modeling and validation
### Assessment of MaxEnt modeling

Maximum entropy (MaxEnt) was used to fit the complex responses to the occurrence data only. Bias files were generated using the Gaussian kernel density of sampling localities tool to increase the weight of presence data points using the SDMToolbox in ArcGIS (*Brown, Bennett & French, 2017*). We used the selected environmental variables and only

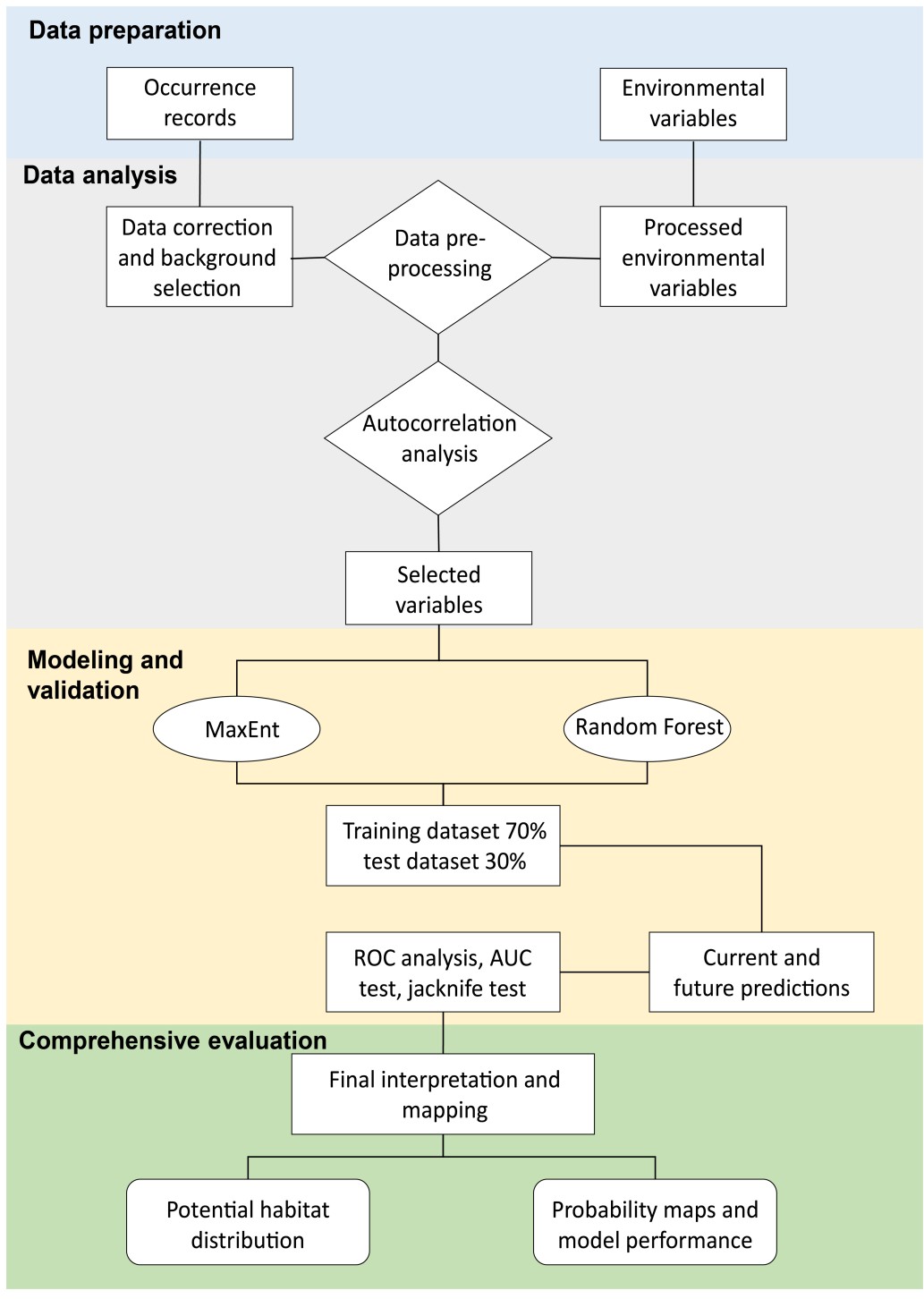

**Figure 2** **Flowchart of the methodology used for the MaxEnt and Random Forest modeling.** Modified from *Liao et al. (2022)*.

the occurrence dataset from MaxEnt v. 3.4.1, implemented in *the kuenm* package (*Cobos et al., 2019b*; *Phillips et al., 2017*) to calibrate the parameter values, evaluate the candidate models, and make future projections. We tested 8,460 candidate models derived from all combinations of three feature classes (linear, quadratic, and product), five regularization multipliers (0.25, 0.50, 1.00, 1.50, and 2.00), and 564 sets of environmental variables (in groups of 6–12 variables). The model sets were trained on 70% of the occurrence data and evaluated on the remaining 30%. The potential distribution of the best model was obtained from the average of 30,000 background points using bootstrap replicates of 500 iterations each, and allowed for free model extrapolation.

Finally, we selected the scenario and candidate models from the "best" set of variables using the selected parameters. To reduce the uncertainty, 10 replicate cross-validation runs were generated to assess the performance of the best model, which was calibrated on a random sample of 70% of the occurrence data and evaluated on the remaining 30% (*Ramos et al., 2019*). The final result was the average of these replicates, and they were used to build the present and future models.

### Random forest modeling assessment

We used the Random Forest (RF) regression algorithm to model the species distribution with discriminatory capacity in the presence and absence of data. We used 10,000 pseudo-absence data with the geographic distance method, assuming that all points were at least 10 km from the presence data to avoid pseudo-absence data because of the dispersal capacity (*Evans et al., 2011*).

We examined the relative importance of each predictor employing the coefficient of determination ($R^2$) in conjunction with the mean squared error (MSE). We implemented a 10-fold cross-validation procedure repeated 10 times to construct RF models with default settings. Notably, the number of trees (5,000), the number of predictor variables at each split (564), and the minimum size of terminal nodes (50) were the key parameters influencing the performance of the RF models.

The dataset, which encompassed both presence and pseudo-absence data, was randomly divided into ten equal subsets. Our modeling strategy involved training the RF model on nine of these subsets and validating it on the remaining subset. It is essential to highlight that we generated 100 RF models and the outcomes were derived by aggregating the results of these models. We used *the randomForest* package (*randomForest* function; *Liaw & Wiener, 2022*) to generate pseudo-absence data and RF models with the selected variables.

## Comprehensive evaluation

We derived the potential distribution from the best model and assessed the relative importance of each variable using the average performance evaluation indicators corresponding to the 'area under curve' (selected models with range of $0.97 < AUC < 1$), and the partial receiver operating characteristic (partial ROC, selected models with a range of $0.90 < ROC < 1$; to determine which of multiple models is most likely to be the best model we used Akaike Information Criterion corrected (AICc; as optimal complexity parameter), and Bayesian Information Criteria (BIC) (*Hirzel et al., 2006*; *Peterson & Soberón, 2012*;

*Jiménez & Soberón, 2020*). A partial ROC was used on 50% occurrence data for bootstrap resampling, 100 iterations, and omission rate error (5%, maximum permissible omission error) (*Cobos et al., 2019b*). Success rate curve were further used to assess the performance of the MaxEnt and RF models in predicting species distributions for model validation (*Rahmati, Pourghasemi & Melesse, 2016*).

Finally, we used Schoener's D index (*Warren, Glor & Turelli, 2008*) to compare the similarity of suitable distribution maps between the MaxEnt and RF models using the *ENMTools* package (*Warren et al., 2021*); this method calculates the random permutation by 100 times the occurrence of *C. angustifolia* in both models. If Schoener's D index is lower (0, no overlap) or higher (1, complete overlap) than 95% of the simulated D values, both models were more dissimilar (or similar) from the suitable distribution than expected by chance (*Sillero, Ribeiro-Silva & Arenas-Castro, 2022*). Maps were averaged to generate spatial information on present and future presence probabilities across all *C. angustifolia*. We followed the ODMAP protocol for the modeling process (Overview, Data, Model, Assessment, and Prediction; Table S1; *Zurell et al., 2020*).

We determined the surface area variation in each climate model (km$^2$) between present and future scenarios (Table 1). The final shapes were obtained using MaxEnt with maps processed using QGIS v. 3.18.3 (*QGIS.org, 2021*). We compared the variation in mean temperature, annual precipitation, and tree loss variation between the present and three future periods using raincloud plots from the packages *ggdist* (*Kay & Wiernik, 2023*), *gghalves* (*Tiedemann, 2022*), and *ggplot2* (*Wickham et al., 2021*). Two-way ANOVA and *post-hoc* Tukey's test were used to compare the means of these variables using the *rstatix* package (*Kassambara, 2023*).

### Predicted refugges to climate change

We concatenated the present and future potential of *Cedrela angustifolia* distribution with predicted refuges to climate change, recognizing appropriate grids in scenarios SSP 3-7.0 and SSP 5-8.5. We used the consensus model between the present and future models, land cover (*Hansen et al., 2013*), and protected natural areas (PNA; *IUCN, 2023*) to recognize and estimate remnant patches outside the PNA. We performed a spatial distribution bias correction to avoid over-adjusting of future projections. We included 10,000 bias files and bioclimatic variables to assess potential refugees for climate change analysis. We implemented a Gaussian kernel analysis using the *kernel* function with QGIS software to avoid sampling bias and identify the highest potential refuges for climate change.

## RESULTS

### Model evaluation and contribution of predictor variables

We generated and compared all candidate MaxEnt and RF models and selected one model from each period that met the criteria of significance, predictive ability, fit and complexity (Table S2). Our results showed excellent performance for both the MaxEnt and RF models when assessed against the independent test dataset. Nonetheless, the MaxEnt model demonstrated a higher AUC ratio; however, RF exhibited better predictive performance than MaxEnt (Table S2).

**Table 2  Percent contribution and permutation importance for present and future models in the two scenarios.** The table shows the periods of time, model names, variables, percent contribution, and permutation importance.

| Variables | Percent contribution in MaxEnt (%) | | | | | | | Increment of node purity in RF | | | | | | |
|---|---|---|---|---|---|---|---|---|---|---|---|---|---|---|
| | PR | 2040 | | 2070 | | 2100 | | PR | 2040 | | 2070 | | 2100 | |
| | | a | b | a | b | a | b | | a | b | a | b | a | b |
| BIO15 | 26.3 | 11.3 | 12.6 | 24.6 | 14.6 | 16.4 | 24.6 | 46.1 | 23.9 | 33.4 | 28.6 | 24.6 | 26.5 | 24.5 |
| SOCS | 20.3 | 21.6 | 15.8 | 15.3 | 15.8 | 17.9 | 26.7 | 40.6 | 36.1 | 29.4 | 34.3 | 37.4 | 37.6 | 32.4 |
| NDVI | 11.7 | 15.3 | 24.3 | 29.3 | 11.3 | 12.3 | 15.9 | 36.8 | 32.2 | 35.7 | 35.7 | 29.4 | 30.5 | 29.1 |
| BIO4 | 8.5 | 4.6 | 13.4 | 14.6 | 16.3 | 6.2 | 11.3 | 34.8 | 34.8 | 35.3 | 29.7 | 40.4 | 37.2 | 29.9 |
| OCD | 7.4 | 7.10 | 8.9 | 11.6 | 15.9 | 4.6 | 14.2 | 28.9 | 20.9 | 21.1 | 20.8 | 19.6 | 21.1 | 18.9 |
| BIO18 | 6.2 | 24.7 | 7.6 | 4.9 | 7.6 | 7.6 | 8.4 | 25.1 | 18.3 | 18.4 | 17.9 | 16.6 | 26.9 | 21.9 |
| SILT | 4.8 | 11.3 | 4.3 | 5.8 | 2.4 | 4.9 | 9.5 | 24.3 | 26.6 | 26.3 | 25.2 | 24.9 | 27.3 | 28.1 |
| CC | 3.7 | 2.8 | 5.6 | 6.7 | 6.4 | 3.8 | 7.6 | 18.1 | 19.3 | 19.7 | 19.1 | 17.1 | 17.6 | 19.4 |
| DEF | 2.4 | 7.6 | 4.9 | 12.6 | 3.9 | 6.4 | 9.3 | 19.1 | 14.3 | 15.3 | 15.6 | 13.7 | 12.7 | 14.3 |
| BIO3 | 2.3 | 4.9 | 2.6 | 7.3 | 6.9 | 3.7 | 6.7 | 14.7 | 20.6 | 18.7 | 19.6 | 21.3 | 17.4 | 18.2 |

**Notes.**

PR, present model;  a, SSP 3-7.0 scenario of future model;  b, SSP 5-8.5 scenario of future model.

For the present and future models, the relative contribution of each predictor variable to the SDMs was assessed by visualizing the percentage contribution and permutation importance (Table 2). In our analysis of the present and future models, we identified ten environmental variables as the most important factors in the model fit. These variables were precipitation seasonality (BIO15), soil organic carbon stock (SOCS), normalized difference vegetation index (NDVI), temperature seasonality (BIO4), organic carbon density (OCD), precipitation of the warmest quarter (BIO18), silt content (SILT), clay content (CC), loss forest (LF), and isothermality (BIO3). These variables showed a high percentage contribution to the model for both the present and future scenarios, as well as for the three periods (Table 2).

## Present potential distribution

The potential distribution with the tree cover loss effect of *C. angustifolia* was approximately 13,080 km$^2$ (mean of MaxEnt and Random Forest values), and Schoener's D index between the RF and MaxEnt models was 0.857. Nevertheless, excluding the effect of tree cover loss, the total distribution was 16,148.5 km$^2$, with a Schoener's D index of 0.749. Notably, the areas in Peru and Bolivia were suitable for *C. angustifolia* according to the current records (Fig. 3).

With the loss forest effect, the distribution detected was 798 km$^2$ in Ecuador, 1,947 km$^2$ in Peru, 9,591 km$^2$ in Bolivia, and 744 km$^2$ in Argentina. Notwithstanding, without loss forest effect, we detected that in Ecuador exhibited 948.5 km$^2$, Peru with 2,579 km$^2$, Bolivia with 11,400.5 km$^2$, and Argentina 1,220.5 km$^2$ (Fig. 2).

We found that the presence of *C. angustifolia* was mainly influenced by these environmental variables: BIO3 (from 5.3 to 6 °C), BIO4 (20 °C to 31 °C), BIO18 (100 to 200 mm), BIO15 (750 to 830 mm), SOCS (0–5%), NDVI (0.95−1.00 units), OCD (0–5%), slit (0–10%), CCF (0–10%), LF (30–55%).

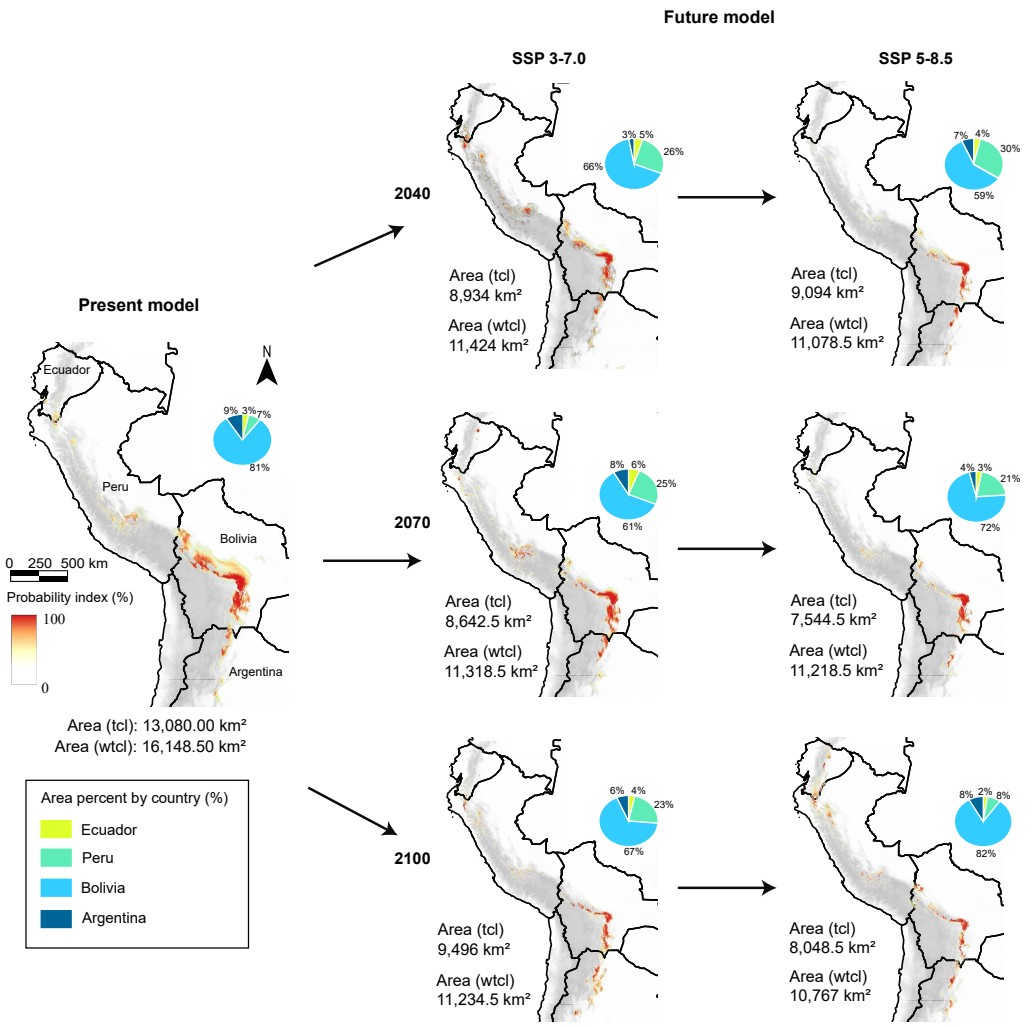

**Figure 3** **Present and future models (SSP 3-7.0 and 5-8.5), and area coverage percentage for each country and model, in present, 2040, 2070, and 2100 periods.** tcl, tree cover loss raster in the model; wtcl, without tree cover loss raster in the model. Realized by Fressia Nathalie Ames Martínez; and base map for countries was obtained from DIVA-GIS (2017).

## Future potential distribution

Under two scenarios, we detected a decrease in the distribution range of SSP 3-7.0 and 5-8.5 during the three periods (Fig. 3). For 2040, we estimated an extension equivalent to 8,934 km² (SSP 3-7.0) and 9,094 km² (SSP 5-8.5), with Schoener's D index of 0.759 (Fig. 4A). Nevertheless, without the effect of tree cover loss, we detected 11,424 km² (SSP 3-7.0) and 11,078 km² (SSP 5-8.5), indicating 29.26% and 31.40% loss forest, respectively, with a Schoener's D index of 0.786 (Fig. 4B).

For 2070, our models predicted a decrease in area from 29.91% (SSP 3-7.0) to 30.53% (SSP 5-8.5) without the loss forest effect, and from 33.93% (SSP 3-7.0) to 42.32% (SSP 5-8.5) with the influence of tree-cover loss, and Schoener's D index was 0.846. Finally, for 2100, the area decreased by 30.43% (SSP 3-7.0) to 33.33% (SSP 5-8.5) without the loss
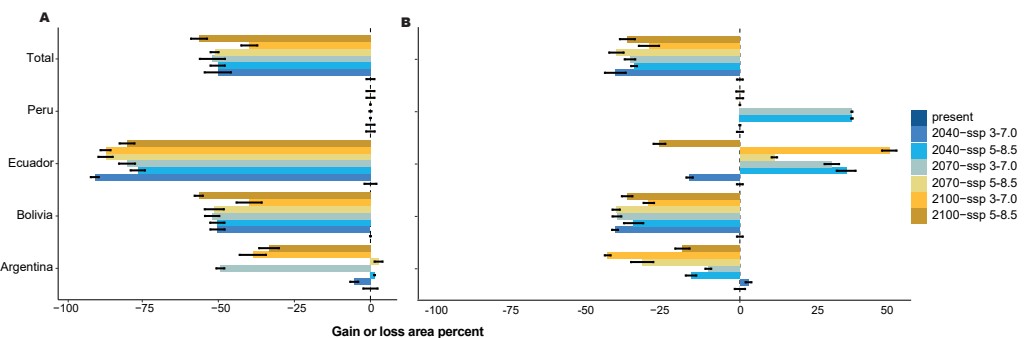

**Figure 4 Gain or loss area percentage for present and future periods, with two scenarios for each country in the present and SSP 3-7.0 and SSP 5-8.5 future scenarios for three periods.** (A) With forest loss effect, (B) without forest loss effect.

forest effect; however, the loss forest effect decreased by 27.40% (SSP 3-7.0) and 38.47% (SSP 5-8.5) in the whole distribution, with Schoener's D index of 0.872 (Figs. 4A, 4B).

The mean temperatures up to 2040 in the two scenarios showed no significant differences among the countries (Figs. 5A–5D). However, Argentina and Bolivia exhibit statistically significant differences by 2070. Similarly, both countries displayed similar annual mean temperature values (~20.5 °C and ~18.7 °C, respectively) for 2040 (Figs. 5A–5B). Ecuador and Peru showed similar annual mean temperature values (~15.2 °C and 16.3 °C, respectively) between the present and 2040 (Figs. 5C–5D). In the four countries, 2100 presented statistically significant differences compared to the other periods (Figs. 5A–5D). In contrast, the annual precipitation showed no significant differences among the four periods or countries (Figs. 5E–5H).

## Effect of forest cover loss in the present and future

The relationships between the predictor and response variables show how forest cover loss affect model predictions. Therefore, we further analyzed how the predicted probability of species occurrence changed with the forest cover loss effect using the marginal responses of the probability of presence suitability (Fig. 6A). We found that 30.59% of the total area was the most suitable habitat without the forest loss; nevertheless, Peru, Ecuador, and Argentina showed variations in the gain of forest loss area in 2040, 2070, and 2100 (Fig. 6B). For example, Ecuador and Argentina decreased by 51.25% and 14.08%, respectively, under the influence of the forest loss effect (Fig. 6B) or the total distribution, and Bolivia increased by more than 30% of the area distribution in all periods, without the forest loss effect. Nevertheless, Peru and Argentina are expected to decrease by 70% by 2070.

## Potential refuges to climate change

Our analysis revealed variations in the habitat suitability distribution areas of *C. angustifolia* (9,449 km$^2$) across the four countries. Nonetheless, Ecuador (724 km$^2$), Peru (1,784 km$^2$), and Argentina (683 km$^2$) exhibited a high potential for refuge from climate change; whereas Bolivia (6,258 km$^2$) displayed a decreasing habitat (Fig. 7). Only 24.28% of the current

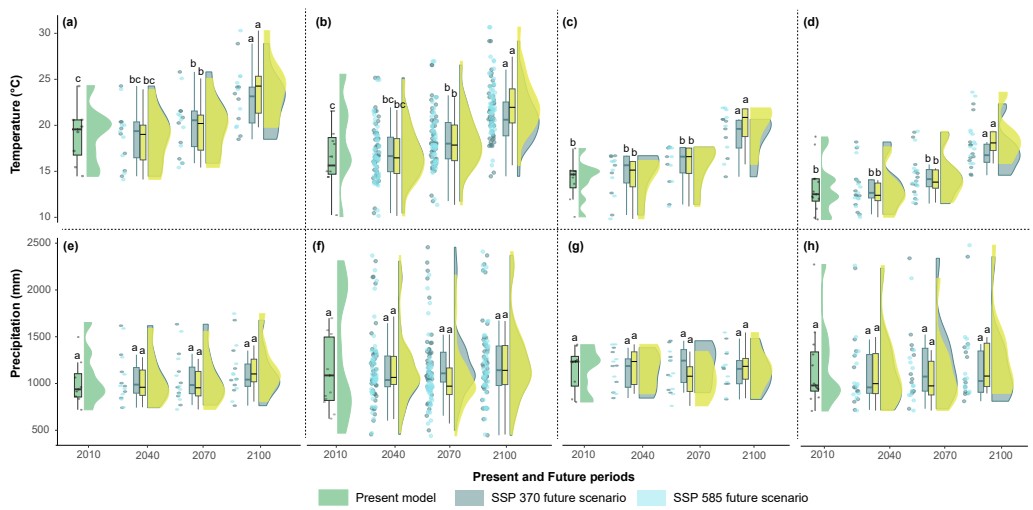

**Figure 5** **Raincloud plots for mean annual temperature (°C) (A, B, C, D) and annual precipitation (mm) (E, F, G, H) (*y*-axis) by present and future periods (*x*-axis) for each country.** (A, E) Argentina, (B, F) Bolivia, (C, G) Ecuador, and (D, H) Peru. The green rainclouds correspond to the present model, the gray rainclouds correspond to the SSP 3-7.0 future scenario, and the light blue rainclouds correspond to the SSP 5-8.5 future scenario. Each rain cloud has a corresponding boxplot (left side). The letters over each boxplot indicate statistically significant ($p < 0.05$) differences between the years 2010, 2040, 2070, and 2100.

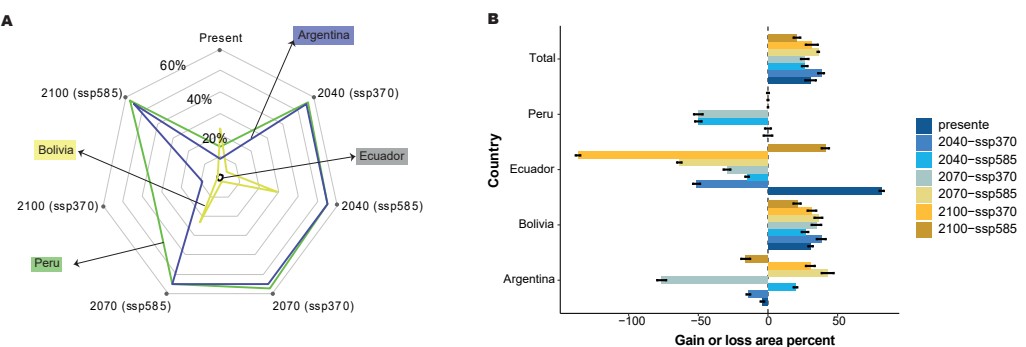

**Figure 6** **Effect of tree cover loss from current to future periods generated for *C. angustifolia*, with two climate change scenarios (SSP 3-7.0 and 5-8.5) for each country studied.** (A) Probability of *C. angustifolia* presence, (B) gain or loss area percentage for each country.

potential distribution is within protected areas, and this is expected to decrease to 25–30% by 2100.

Predictions of the core distribution regions from the MaxEnt and RF models showed a higher heterogeneity and stronger gradients. In addition, our analysis identified Azuay and Zamora Cinchipe as refugees in Ecuador; Cajamarca, Junín, Apurimac, and Cusco as refugees in Peru; Cochabamba, Chuquisaca, and Tarija as refugees in Bolivia; and Salta and Tucumán in Argentina as potential refugees to climate change.

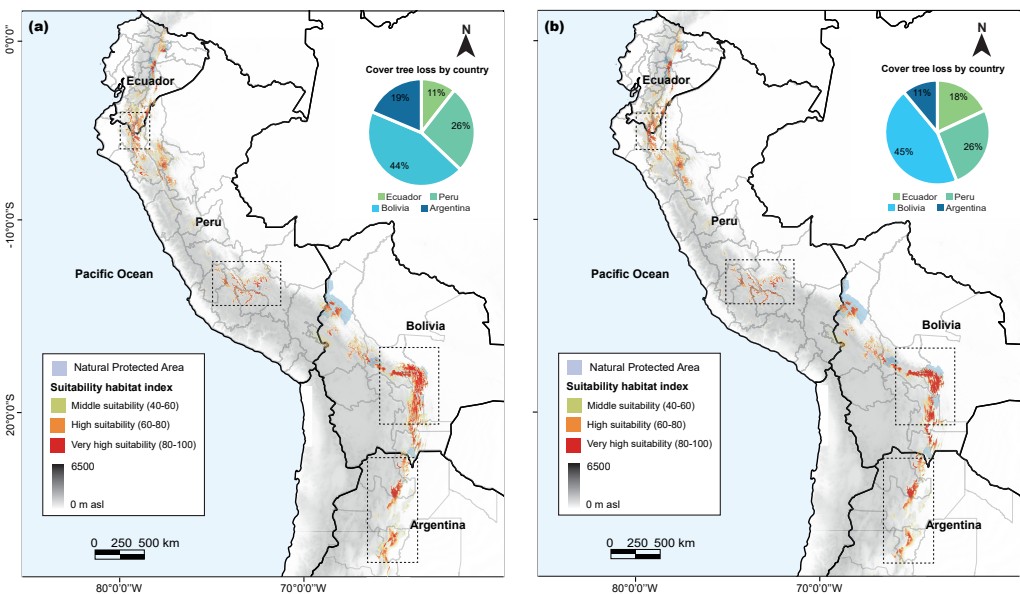

**Figure 7** *Cedrela angustifolia* suitability habitat by country and for specific protected areas in the diferent countries, under combined of present and future climate change scenarios, cover tree loss percent for two algorithms and areas with most change. (A) MaxEnt model (B) Random Forest model. Both models made by Fressia Nathalie Ames Martínez and base map for countries was obtained from DIVA-GIS (2017).

## DISCUSSIONS

The effects of climate change on Andean montane forests are well-known, but the effects on specific pioneer relict tree species are unclear. Our findings suggest that the distribution of *C. angustifolia* will decrease in the future owing to climate change projections for 2040, 2070, and 2100. During these time periods, high temperatures, low precipitation and high $CO_2$ concentrations significantly affected the distribution of the species (*Eyring et al., 2016*).

### Present potential climate sensitivity

Previous studies have systematically evaluated the predictive effectiveness of RF and MaxEnt models through automated parameter optimization (*Cotrina et al., 2021*; *Mi et al., 2017*; *Zhao et al., 2022*). Although the RF models typically provide robust and accurate predictions using default configurations (*Freeman et al., 2015*), MaxEnt models often require parameter refinement (*Feng et al., 2019*; *Jiménez & Soberón, 2020*). We selected the best feature classes and regularization parameters for MaxEnt and compared them with the RF model. Consistent with previous literature, both the RF and optimized MaxEnt models exhibited commendable predictions accuracies; however, the RF model showed a marginal superiority, which was evident in both cross-validation and external dataset evaluations. This is consistent with the findings of *Mi et al. (2017)* and *Čengić et al. (2020)*, who described the improved performance of RF over standard MaxEnt configurations for species distribution prediction.

The distribution of *Cedrela angustifolia* is influenced by seasonal variations in precipitation and temperature, and the model performance is excellent (AUC > 0.98), demonstrating the efficiency and accuracy of the model (*Warren & Seifert, 2011*). These populations exhibit tolerance to colder temperatures and higher moisture conditions, maintaining their evolutionary climatic conditions, as detected by *Muellner et al. (2010)* and *Koecke et al. (2013)*. Our analysis suggests that Bolivian montane forests provide a suitable ecological assemblage (81% present model), and ecosystem conservation for *C. angustifolia* (*Pennington & Muellner, 2010*), with more than half of the records of this species within the NPAs.

## Effect of climate change in the future

The distribution patterns of different species are influenced by different ecological and evolutionary factors that allow them to survive in specific environments within diverse landscapes (*Rahbek et al., 2019*). Despite this, *Tejedor Garavito et al. (2015)* argued that land use change, particularly deforestation of AMFs, would be more detrimental to biodiversity than climate change, leading to the loss of relict-endemic tree species worldwide (*Feeley & Silman, 2010*).

Based on the evaluated scenarios, a temperature increase of 4.1 °C to 5 °C is expected compared the present temperature record. This would result in a significant reduction in suitable habitat in Bolivia (>20.26%) and Argentina (>28.99%). By 2100, *C. angustifolia* will not be able to find optimal sites, because the thermal limit will be exceeded (∼21.8 °C). However, Peru and Ecuador offer favorable climatic conditions that would allow *C. angustifolia* to migrate to cooler areas and persist through rapid changes in climate (*Pearson, 2006*). This confirms the climatic impact of *C. angustifolia* (*Cotrina et al., 2021*; *Koecke et al., 2013*; *Rodríguez-Ramírez et al., 2022*). Similar results have been reported for *C. odorata* (*Sampayo-Maldonado et al., 2023*) and other *Cedrela* species (*Cotrina et al., 2021*; *Koecke et al., 2013*).

We demonstrated that tree cover loss affects more than 30% of the range of *C. angustifolia*, which is a critical factor contributing to the reduction in *C. angustifolia* in the four countries. If the current rates of deforestation continue or increase, this will lead to reduction in distribution (*Hansen et al., 2013*). Similarly, the loss of tree cover increases the risk of soil erosion, leading to reduced soil fertility and increased sedimentation in the water bodies in these forests (*Bax & Francesconi, 2018*; *Cuenca, Arriagada & Echeverría, 2016*; *Tapia-Armijos et al., 2015*). Therefore, the climatic impact of deforestation on AMF weakens the relationship between atmospheric circulation and the hydrological cycle (*Longobardi et al., 2016*).

## Potential habitat suitability and refuges

Suitable habitat for *C. angustifolia* will be maintained in all four countries; however, habitat suitability will decrease in all countries in contrast to the present and future models evaluated. Bolivia exhibited more illegal logging and forest fires than the other three countries, making it more vulnerable to climate change (*Hansen et al., 2013*). In contrast, Ecuador, Peru, and Argentina had areas with better habitat suitability than the

present model, suggesting that unexplored forests with similar climatic conditions allow the species to adapt through natural or active restoration (*i.e., C. angustifolia* plantations in inter-Andean valleys; *SERFOR, 2020*).

Furthermore, their presence increases in NPAs, as described by *Pennington & Muellner (2010)*, which shows that NPAs function as biodiversity reserves and buffers against the effects of changing climatic conditions, allowing the formation of refugia and providing ecological corridors for species to acclimate or migrate over the long term (*Cuesta, Peralvo & Valarezo, 2009*; *Geldmann et al., 2013*). In contrast, 75.72% of habitat suitability was detected outside the NPAs, indicating the need to develop forest management and monitoring strategies to protect these forests as they are more susceptible to selective logging and timber overexploitation, as we demonstrated by our results (*Cotrina et al., 2021*; *SERFOR, 2020*).

## CONCLUSIONS

Our study demonstrated that *Cedrela angustifolia* is vulnerable to future climate change, indicating differences in suitable habitats between Central and South America. Moreover, we propose the designation of a climate sanctuary for *C. angustifolia* associated with NPAs and land-use changes. It is therefore crucial to collaborate with local communities living near forests to protect endangered and vulnerable CITES and IUCN species and their habitats, both inside and outside NPAs.

*C. angustifolia* is ecologically important because it provides habitat and food for species, prevents soil erosion and promotes sustainable management. In addition to its value as timber, the extinction of *C. angustifolia* also has a profound impact on the environment. Its loss would lead to reduction in biodiversity, as many organisms depend on it for food and habitat. The disruption of these relationships can destabilise ecosystems, affecting other species and leading to cascading effects. Moreover, the disappearance of *C. angustifolia* could undermine the cultural heritage of indigenous communities who depend on it for their livelihoods and traditions. Trees also play a role in carbon sequestration and contribute to climate regulation. Therefore, preserving *C. angustifolia* is crucial not only for ecological balance, but also for maintaining cultural diversity and combating climate change.

Non-governmental organizations (NGOs) and several environmental legacy institutions are using the essential findings and boundaries established by the Species Distribution Model (SDM), both now and in the future, to safeguard and preserve the species under investigation. Therefore, it is crucial to emphasize the outcomes of our study, and consider the need to initiate conservation efforts for *C. angustifolia* (as an umbrella species), including the establishment of new protected areas, habitat restoration, and the creation of ecological corridors that benefit other related species.

Combating deforestation and climate change in the Andes requires coordinated efforts at local, national and international levels. The distinctive ecosystems of the AMF and the well-being of its inhabitants must be preserved through the implementation of conservation programs, sustainable land-use plans and climate change mitigation initiatives. To ensure

the effective regulation of *C. angustifolia* logging, propagation and restoration programs, it is crucial to help local authorities in understand the ecological significance of these practices. Furthermore, we suggest that additional research on other aspects, such as phenology, functional ecology and spatio-temporal patterns, be conducted to gain a deeper understanding of how tree species in the Andean montane forest (AMF) are responding to the impacts of climate change and human activities.

## ACKNOWLEDGEMENTS

We thank the Servicio Nacional Forestal y de Fauna Silvestre-SERFOR, which approved the research outside protected natural areas through the General Management Resolution RDG 007-2020-MINAGRI-SERFOR-DGGSPFFS.

### Funding

This research was supported by Consejo Nacional de Ciencia, Tecnología e Innovación Tecnológica-CONCYTEC, under Grant 086-2018-FONDECYT-BM-IADT-SE. The funders had no role in study design, data collection and analysis, decision to publish, or preparation of the manuscript.

### Grant Disclosures

The following grant information was disclosed by the authors:
Consejo Nacional de Ciencia, Tecnología e Innovación Tecnológica-CONCYTEC: 086-2018-FONDECYT-BM-IADT-SE.

### Competing Interests

The authors declare there are no competing interests.

### Author Contributions

- Fressia N. Ames-Martínez conceived and designed the experiments, performed the experiments, analyzed the data, prepared figures and/or tables, authored or reviewed drafts of the article, and approved the final draft.
- Ivan Capcha Romero performed the experiments, analyzed the data, authored or reviewed drafts of the article, methodology, and approved the final draft.
- Anthony Guerra performed the experiments, authored or reviewed drafts of the article, methodology, and approved the final draft.
- Janet Gaby Inga Guillen performed the experiments, authored or reviewed drafts of the article, methodology, and approved the final draft.
- Harold Rusbelth Quispe-Melgar performed the experiments, authored or reviewed drafts of the article, and approved the final draft.
- Esteban Galeano performed the experiments, authored or reviewed drafts of the article, and approved the final draft.

- Ernesto C. Rodríguez-Ramírez conceived and designed the experiments, performed the experiments, prepared figures and/or tables, authored or reviewed drafts of the article, and approved the final draft.

## Data Availability

The raw data is available in the Supplementary File.

## Supplemental Information

Supplemental information for this article can be found online at http://dx.doi.org/10.7717/peerj.18799#supplemental-information.

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
