# Peer review of "Climate change and tree cover loss affect the habitat suitability of Cedrela angustifolia: evaluating climate vulnerability and conservation in Andean montane forests"

_PeerJ, doi:10.7717/peerj.18799_

## Round 0.1 · original submission · Major Revisions

The reviewers have identified major edits that need to be made. These include language edits and clarity in methods. In addition, the model needs to be revisited to address some of the comments. I suggest carefully reviewing the comments and pursuing a major paper revision.

**Language Note:** The Academic Editor has identified that the English language must be improved. PeerJ can provide language editing services - please contact us at [email protected] for pricing (be sure to provide your manuscript number and title). Alternatively, you should make your own arrangements to improve the language quality and provide details in your response letter. – PeerJ Staff

Reviewer 1 ·

Basic reporting

The manuscript would benefit from a careful and thorough grammatical review. The sentence structure is frequently missing fundamental components including a subject, verb, or object. Such issues make it difficult to assess the methods, results, and conclusions of the study.

Experimental design

The objectives of the study are clear and within the aims and scope of the journal. However, the methods are incomplete, confusing, and insufficiently described to properly assess their suitability and ensure that all results are reproducible.

Validity of the findings

It is difficult to properly assess the validity of the findings without first understanding the methods applied and how they apply to the specific questions/objectives of the study.

Annotated reviews are not available for download in order to protect the identity of reviewers who chose to remain anonymous.

·

Basic reporting

The organization and language of this paper is unclear and often confusing, with many grammatical mistakes (please see attached copy with specific suggestions). One concern is that important terms need to be defined/explained (ex. L67 - "relict-tree populations", L78 - "relict species", L346 - "pioneer relict species", L124 – “climate sensitivity”) while the use of the term "thermophilization" is incorrect and incorrectly cited. Citations need to be added to the first sentence of the discussion (L344-345) to place the sentence in context. The framing of the hypothesis (begins L115) makes it sound like they "identified the effectiveness of protected areas and suitable areas for restoration..." but I don't think this was actually done (or if it was it was not made clear what those areas were in the results/discussion). Additionally, the discussion references the role of CO2 concentrations (L349 and L390-394) but I do not believe this was explicitly tested in the model. I also think the discussion could use a paragraph on the ecological importance of Cedrela in the forests and why we should care about its future as shown in their models.

Experimental design

This is an original primary article that fits well within the scope of PeerJ. Hypothesis/research questions are not clearly defined (please see my previous comments on the section beginning L115), but it is relevant and meaningful and fills and important knowledge gap for this species. I believe this to meet the requirements of a rigorous investigation, but much of the methods is unclear/requires more explanation. In the current state I don’t believe the methods to be in sufficient detail for replication, but this is easily corrected. Most significantly, I don’t believe the selection of the bioclimatic variables to be sufficiently explained, and the paragraph that supposedly lists the 19 variables (begins L186), only includes 16. Some specific concerns on the methods below, but please see the attached document.

L134 – why was the study area truncated to 1800-3300m?
L144 – how did you decide points were “incorrect” or “outliers”?
L145 – please explain more on using “geographic distances to narrow datasets”, this was unclear.

Validity of the findings

I believe the underlying data to be robust and statistically sound. However, I am confused why the first “future” period they modeled was 2011-2040, when more than 1/3 of this period is already in the past. Instead, it seems like the 2011-2024 period would be an interesting test of the predictive ability of their model, and changes that have occurred during this period important for their conclusions. It seems odd for the “present model” (Fig 2) to represent pre-2011, especially when it looks like data was collected up to 2022. The x-axes in Fig 3 are misleading (and actually obscures some of the effect) because different scaling is used between panels A and B and makes them look more similar than they are. In Fig 4 I cannot differentiate between the two blue colors. The conclusions in the discussion about CO2 concentrations are not supported by the study (L390-394).

---

## Round 0.2 · Major Revisions

There are major issues that still need to be addressed. Please review the reviewers' comments, address each, and justify your responses. The writing needs to be clearer in several places that need to be addressed. The experimental design and methods need to be revised for more clarity and information. Validation also needs to be addressed and written with clarity.

Reviewer 1 ·

Basic reporting

The authors have made an observable effort to improve the clarity of the writing and fix any grammatical issues. However, I still find the writing to be consistently unclear with many examples of incomplete sentences, ambiguous wording, and confusing paragraph structure. Some of these issues may be fixed by a careful grammatical review, however others suggest the lack of clear ecological story being conveyed by the authors.

Specific examples of ambiguities where the point that is being conveyed in one or more sentences is unclear or does not follow from previous sentences include (this is not an exhaustive list): lines 68-69, 72-73, 101-102, 115-118, 136 (what does 3-900 km define?), 138-142, 144-145, 153-155, 298-300.

Other comments related to basic reporting:

Table 1 is confusing in its current form. I cannot tell what variables were included in each model, nor what were the different models included in the study.

PA is used in the abstract without definition (I assume this means protected area?).

Line 88 states Cedrela is a genus, then later states it is a species.

Lines 121-128 It is unclear how study objective (3) is different from (1).

Line 152 states 10 soil rasters were used, but line 172 states 11 were used?

Line 219 It is not clear what “scenario” refers to here.

Experimental design

While the authors have revised their description of the methods used in the analysis, I still find that the methods are not sufficiently described to understand and evaluate the modeling approach and the study findings. Specific questions, comments, and examples are provided below.

In my previous review, I commented that citing R packages (or comparable) is not a sufficient description of applied methods and that the authors need to describe the methods that were applied. In the revised manuscript, the authors have added the specific functions that were applied within such packages, but do not describe the methods. For example, on lines 182-183 the authors state they use the “resample” function with the raster package to correct the spatial misalignment in the spatial data inputs. However, a description of the method applied by the resample function to correct misalignment is not provided. The reader should not have to go to the help page of an R function to determine the methods used in the study. I picked this particular example because the misalignment of the inputs has the potential to greatly impact the modeling results and, as such, the approach taken to correct it is important…there are many other examples of citing functions or studies in lieu of describing methods throughout the manuscript that need to be addressed. Make it easy on the reader to understand what was done.

It is unclear if tree cover loss is based on NDVI observations (it seems that it is) or is a parameter of the future emissions scenarios. If it is based on the former, then there are no future observations of tree cover loss (observations are available only through 2022 per Table 1). If so, then how was tree cover loss considered in future predictions. Based on lines 202-204, it seems that NDVI was included as a predictor in some forecasts, but not others. However, this compares the current levels of tree cover loss (from 2022) to a scenario where there is no tree cover loss. This is not the same as accounting for future tree cover loss, which based on the introduction and discussion, seems likely to increase in the future. This is important to properly describe as it relates to the objectives and main findings of the study.

The cross-validation (CV) strategy is not properly described. It seems that one CV strategy was used to assess the MaxEnt model (lines 206-22) and another CV strategy was used for the RF model (lines 224-239). Yet under the comprehensive assessment, the predictions of both models to out-of-sample data are described and evaluating suggesting that the same CV strategy was used. Which is it? Why were different CV strategies applied? The description of CV within a model is also confusing. For example, under the MaxEnt model the authors state that 70% of data were used for training and 30% for testing. This is not CV which iterates training and testing the model on multiple subsets of data, but rather a single holdout set. Then below, the methods state that 10 replicate runs of CV were applied. Do you mean replicates of the single fold described above or 10-fold CV? The same is true for the RF model…on line 230 the methods state that 100-fold CV was applied, but on lines 234-235 they state that the data was partitioned into ten equal subsets implying 10-fold (not 100-fold) CV?

The comprehensive evaluation subsection is not clear and does not provide sufficient detail to understand what was done. For example, from lines 254-255 I have no idea what thresholding strategy was used to convert predictions of species probabilities to binary data. What is a “logistic threshold”?

Validity of the findings

Given the lack of sufficient detail in the methods, I do have sufficient information to properly assess the modeling approach applied or the validity of the results presented. In general, there appears to be a disconnect between what is presented in the results and primary discussion points. That is, the discussion focuses on specific findings that are not highlighted in the results. The authors need to do a better job calling the readers attention to the key findings in the results section before detailing these within the discussion. Make it easy on the reader to find the important information.

·

Basic reporting

The writing was greatly improved in this version of the manuscript, however I still find instances of grammatical errors or where clarity is lacking throughout. I highlighted some examples in the text (ex. title, lines 49, 51, 71-72, 76, 94, 128-130, 307, 346, 433) but there are others which hinder communication. References and background information are sufficient, though I still think this paper would benefit from a stronger and more direct argument for why C. angustifolia is an important species and why predicting and protecting its range is important (in both the introduction and conclusion). My issues with the hypotheses have been corrected.

Experimental design

This is an original primary article that fits well within the scope of PeerJ. As mentioned, the hypotheses/research questions have been clarified and now match with the reported results. The methods section has been significantly expanded and are now of sufficient detail and clear (aided by the addition of figure 2). My biggest concerns on the experimental design have been addressed (please still see the attached document for minor comments in section 1.1 and 1.2), though the specifics of these SDMs are not my area of expertise and comments from reviewer 1 should be prioritized.

Validity of the findings

The changes to the manuscript surrounding the modeling periods greatly improved the clarity and addressed my concerns. However, I still find issue with some of the conclusions in the paragraphs beginning on lines 441 and 453. In the paragraph beginning on 441, you do not demonstrate that this species has the potential to adapt or acclimate to climate change or even that it can migrate. Only that if it is able to migrate fast enough, there is potential habitat available for it. Additionally, in the paragraph beginning on line 453, I still don’t understand the connection of the SDMs to low CO2 absorption and would remove these sentences entirely as it only confuses the message. The updated figures address my concerns and are now clear.

---

## Round 0.3 · Minor Revisions

Please check for grammatical errors. Explain all methods clearly instead of referring to R packages. Provide details on statistical methods.

Reviewer 1 ·

Basic reporting

The writing in the revised manuscript is much improved and is sufficiently clear despite some lingering grammatical issues.

Experimental design

The authors have improved the clarity and detail provided in the methods section. I have a better understanding of the methods applied. I still feel that the manuscript does a poor job of fully explaining the methodologies often citing R packages and functions in lieu of describing methods.

Validity of the findings

Findings are more clearly reported with the discussion following more closely from the results. This is significantly improved from the last version of the manuscript.

Additional comments

No additional comments at this time.

---

## Round 0.4 · accepted · Accept

The manuscript is improved.